# Home Monitoring of Oxygen Saturation Using a Low-Cost Wearable Device with Haptic Feedback to Improve Sleep Quality in a Lung Cancer Patient: A Case Report

**DOI:** 10.3390/geriatrics7020043

**Published:** 2022-03-31

**Authors:** Walter Lachenmeier, Dirk W. Lachenmeier

**Affiliations:** 1Deutsche Forschungsgemeinschaft (DFG, German Research Foundation), 53170 Bonn, Germany; walter.lachenmeier@web.de; 2Chemisches und Veterinäruntersuchungsamt (CVUA) Karlsruhe, Weissenburger Strasse 3, 76187 Karlsruhe, Germany

**Keywords:** wearable sensor, pulse oximetry, sleep disturbance, blood oxygenation, haptic feedback, home care, oxygen concentration

## Abstract

This study reports the case of a lung cancer patient with increasing difficulties in falling asleep and frequent periods of wakefulness. Severe dyspnea related to pneumonitis caused as a side effect of immunotherapy worsened the situation. Eventually, a fear of falling asleep developed, including panic attacks and anxiety around choking, which was shown to lead to nights of complete wakefulness. The patient did not only sleep poorly; he did not sleep at all at night for several days, as evidenced by the notes he made during the night. Polygraphy showed no evidence of sleep-disordered breathing, but frequent periods of wakefulness and a reduced basal saturation of around 90% during sleep due to lung changes such as an extensive functional failure of the left upper lobe with position-dependent shunts. The authors hypothesized that the symptoms described were causally related to a drop in oxygen saturation in the patient’s blood. Therefore, they pursued the goal of finding a measurement technique that is as inexpensive as possible and that the patient can operate without outside assistance and great effort. Thus, the patient started using a low-cost wearable device that allows simultaneous measurements of blood oxygen content, pulse rate, and movement intensity. It consists of a finger ring with a pulse oximetry sensor and a wristband with a control unit containing a vibration motor. The described device reliably warned of disturbances in the oxygen concentration in the blood during the night with its vibration alarm. By use of that device during the whole night at home, the events of reduced oxygen saturation and anxiety symptoms were reduced. Sleep disturbances with sudden awakenings did not occur when using the device. The patient benefited from the security gained in this way and slept much more peacefully, and he could spend nights without waking up again. In conclusion, wearable oximeters with vibration alarms can be recommended for patients’ home care in lung cancer patients.

## 1. Introduction

Over the last decade, there has been a considerable increase in interest and research activity in wearable sensors to measure blood oxygen saturation (SpO_2_) [1,2,3,4,5,6,7,8,9].

Devices are typically constructed as ring-type sensors, which are controlled using cabled or wireless control units, often with the possibility of connecting to a desktop computer or smartphone using a USB cable or Bluetooth technology [1,3]. Other approaches have suggested the use of 4G, WiFi, or Zigbee networks to upload data to a server, which could be accessed by remote coaches or doctors [4,8]. During a validation study against polysomnography in a sleep center, overnight pulse oximetry in a home setting was found to provide a satisfactory diagnostic performance in detecting severe obstructive sleep apnea [5]. The principle of pulse oximetry is typically applied, which assesses the level of blood oxygenation by optical light transmission through the blood (for a review of oximetry techniques and their theory, see Nitzan et al. [9]).

Another innovation was the combination of SpO_2_ sensors with some form of haptic feedback system, such as an embedded vibration motor [6]. For example, Forra Wakidi et al. suggested a wearable device even in a clinical setting to supervise premature babies. The photodiode sensor combined with a vibrating motor was successfully applied to stimulate the baby’s body when apnea occurs, as well as to raise an alarm at the nurses’ station [2]. The usefulness of such a system to alert patients with obstructive sleep apnea was also described by Brugarolas et al. [7].

The focus of this medical case report was the point-of-care application of an easy-to-use and simple-as-possible wearable oximetry device with haptic feedback, with the objective of helping a geriatric lung cancer patient to handle his severe sleep disturbances.

## 2. Case Report and Methods

### 2.1. Case Description

A 76-year-old man with shortness of breath, productive cough, chest pain, fatigue, and a history of hypertension was diagnosed in November 2020 with non-small cell lung carcinoma (UICC8 classification IIIB). The patient was a smoker with a history of 300 packs per year but had stopped smoking two years before diagnosis. Curative radiochemotherapy with carboplatin and paclitaxel was conducted between December 2020 and January 2021, leading to stabilization. A biopsy of the remaining compression in the upper lobe showed no malignancy, with only inflammatory tissue due to radiation. Bi-monthly immunotherapy with durvalumab was initiated in February 2021. Even before the switch to durvalumab, increasing dyspnea was observed and systemic steroid therapy with 30–40 mg was started. Progressive dyspnea on exertion was reported during the following period, as well as insomnia under immunotherapy and panic attacks. The patient reported a shortness of breath, especially when lying down, so that he used to sleep while sitting at a desk with his head on pillows. In July 2021, a pneumological diagnosis using polygraphy showed no evidence of sleep-disordered breathing, but frequent periods of wakefulness and a reduced basal saturation of around 90% during sleep without oxygen due to lung changes. A blood gas analysis under stress showed no respiratory insufficiency, no ventilatory insufficiency, and a balanced pH. Part of the shortness of breath was judged to be pulmonary, certainly explained by the extensive functional failure of the left upper lobe with position-dependent shunts and a narrow connection to the lower lobe. Polysomnography was not conducted for cost-benefit reasons and because the patient declined another hospital stay at a sleep lab. In August 2021, inpatient treatment was necessary following durvalumab infusion because a normocytic normochromic anemia required transfusion for diffuse small intestinal bleeding developed in the patient. The patient was discharged in a stable general condition after hospitalization. As the causes of the sleep disturbances were psychooncologically determined to be at least partially due to anxiety disorders, including panic attacks and an anxiety around choking, the patient started using the wearable device described in Section 2.2 in August 2021. However, in September 2021, the severity of the dyspnea increased and pneumonitis occurred following resumed immunotherapy leading to further hospitalization. Symptomatic therapy, including oxygen and steroid therapy, stabilized the patient. After discharge, it was decided to abort immunotherapy, which was considered a possible cause of the pneumonitis and previous anemia due to non-specific autoimmune responses. Subsequently, the patient considerably improved and was able to sleep through the night. Fatigue, shortness of breath, and coughing were reduced, and the patient remained stable for a 3-month follow-up period without the need for oxygen ventilation.

### 2.2. Materials and Methods

The SleepU device (Wellue Health, Shenzhen Viatom Technology Co., Ltd., Shenzhen, China) was applied. It allows simultaneous measurements of blood oxygen content, pulse rate, and movement intensity. It consists of a finger ring with pulse oximetry sensors and a wristband with a control unit containing a vibration motor. The ring is connected to the control unit using a USB cable. The USB port is also used for recharging the device, which is necessary every two days. The specified measurement ranges are 70–99% (oxygen level) and 30–250 bpm (pulse). The measurement interval is 15/min. The advantages of this device include the intermediate storage of data, their graphical output on a smartphone, and the possibility of exporting them as a PDF file or CSV file. The ViHealth App for Android v.2.72.0 (Shenzhen Viatom Technology Co., Ltd., Shenzhen, China) was used for data access and export. For a more detailed analysis of the data, Microsoft Excel version 2019 was applied. The device records #255 and #65535 when no reading is available, due to circumstances such as movement or bad signal (i.e., these are numerical values generated by the system to be recognized as error). These readings were excluded and only valid numbers were used (similar to the approach described by Muratyan et al. [10]). Figure 1 shows the device and application. Sleep experiments of relatively short and long durations (nap and night sleep) were initially performed without oxygen supply. Subsequently, such experiments occurred with a manually initiated oxygen supply from a stationary device Type 525 DS from deVilbiss (Somerset, PA, USA). The SleepU device remained switched on throughout. Figure 1 shows a picture of the device and the application.

## 3. Results

Figure 2 shows as a blue curve the course of oxygen saturation in the blood of the patient who was acutely ill with pneumonitis. In this extreme example, the threshold value of 85% (blue dashed straight line) was undershot by at least 1% 26 times between 23:00 and about 4:00. In most cases, the patient awoke during the deficiency phases, which lasted up to 48 s. The exact times for falling below the threshold by at least 1% are shown in Table 1. There was no oxygen supply during the period, up to about 4:00 a.m. Oxygen was supplied to the patient only after that. As the blue curve shows, the oxygen saturation then increased significantly within a few seconds and there were no further drops below the threshold value. The pulse per minute of the patient is shown as a green curve, which demonstrates values in the sleep phases of just below 80/min, and in the movement phases slightly above 80/min.

The values of the red curve are a measure of the intensity of the movement. The numerical values for the movement (arbitrary units) are not quantified in more detail, but are also not necessary for this investigation. Strong fluctuations in the oxygen supply are coupled to phases of intense movement.

For the experiment shown in Figure 2, the duration of the phases with a 1% fall below the threshold and the corresponding start and end times are given in Table 1. The vibration alarm was set to respond at 85% oxygen saturation. Without the alarm, these undershoots are associated with sudden awakenings and lead to periods of wakefulness if the deficiency is prolonged. The results shown are typical of several other experiments that showed similar results.

Figure 3 shows sleep phases first without, then with, oxygen supply to the patient for a sleep of about 1.5 h duration. At the beginning, an awake phase can be observed, with a fall below the threshold value of longer duration, which is terminated by the patient moving around and breathing more intensively. After the patient falls asleep, further undershoots occur, so that the vibration alarm is triggered and the patient wakes up. The corresponding numerical values are summarized in Table 2.

With the oxygen supply starting from approximately 13:00, a significantly higher and more balanced profile of the oxygen content in the patient’s blood can be observed.

The course of the oxygen concentration in the blood of the almost-recovered patient without oxygen supply during the night sleep is shown in Figure 4, next to the pulse course and movement profile. Here, only a single oxygen undersupply occurred (Table 3), whereby the vibration alarm reliably awakened the patient. This event lasted 28 s and reached a minimum value of 82%. The threshold value of 85% was reached again later but no value below this was observed. According to experience, the detected undersupply occurs within about 10 s, starting from a still sufficient oxygen saturation in the range of about 90%. The sensor thus works fast enough to trigger the alarm on time. It is noticeable that the patient’s pulse drops to values significantly below 80/min compared to the case of acute pneumonitis (cf. Figure 2).

## 4. Discussion

The oxygen concentration in the blood is subject to fluctuations in lung cancer patients. The time period of falling below a given threshold value of oxygen concentration was determined. There is no uniform information or agreement in the literature on the level and pathophysiological significance of threshold values for an oxygen shortage, specifically in acutely ill oncologic patients [11,12,13,14]. For this work, it was set at 85% because the patient studied noted a clear relationship to his sleep disturbances in several trials at this value.

He suffered from frequent sudden awakenings from deep sleep or after nightmares. This disturbance progressed to panic attacks over a few weeks. The fear of such attacks later also led to difficulties in falling asleep. It was determined that these problems with sleeping through the night clearly correlate with an undersupply of oxygen in the blood (hypoxemia). On the basis of normal values, the aforementioned threshold value of 85% is regularly reached within a period of a few seconds. A subsequent undersupply in this range lasts on average between 8 and about 40 s. The patient awakens within 4 s after the vibration alarm is triggered and can take countermeasures, such as increasing respiratory intensity. Thus, values above 90% are reached again within a maximum of 40 s; only in a few cases was more time required (Table 1). Thus, the evaluated device seems to be well suited to provide patients with sufficient protection and a sense of security during sleep. The advantages of the device include that the data can be transferred easily in the form of PDF reports or CSV files for further evaluations. For this purpose, simple Microsoft Excel evaluations, for example, can be used to selectively view the records in the area of particular interest near the threshold from several thousand rows.

When using the device, the patient will learn from these data which threshold is the most appropriate for his individual situation. This means he can determine at which oxygen concentration he starts to move or wakes up and derive from this his individual threshold for the vibration alarm. This suggestion could be a future research topic to implement algorithms to suggest an individual threshold value with a few parameters typical for the patient. However, if one wants to do this calculation completely exactly, one would have to assign a caregiver to it, who observes the patient exactly, such as in a sleep lab. Nevertheless, the device covers a lot of what happens in the sleep lab. Patients with severe sleep apnea have up to 100 breathing episodes per night [12]. However, the measurement technique used did not and should not clarify whether these or other possible causes of oxygen deficiency were present.

One advantage is clearly the capacity for home use. The patient does not need to go to a sleep lab for one night, which induces tremendous stress, especially for geriatric patients, since the measurements can be done at home. Then, the report can be sent to the physician by email who can make the necessary conclusions, such as the need for oxygen supplementation, or make suggestions for setting the threshold value for the vibration alarm.

The crucial advantage of the device is the security it gives via the vibration alarm. The patient does not have to be afraid any longer that their breathing will stop while asleep. Especially at the onset of the disease, the patient under study became afraid of falling asleep and became scared and startled when noticing decreased breathing. This effect has completely disappeared because of the device. Hence, the device is especially advantageous for patients, for which sleeping problems may be due to psychological rather than somatic pathological reasons.

An interesting further application of the same device was published by Muratyan et al. [10]. The data of the device were used to identify the individual user with high accuracy.

Further developments could focus on directly coupling the device to oxygen supply systems, e.g., using the Bluetooth connection that is already built in. Thus, instead of the vibration alarm, the oxygen supply could be switched on.

In the future, the applicability and reliability of the device could be improved by including more vital parameters to better guide its decisions, for example, respiratory rate and ECG.

A future feature worth investigating would also be the possibility of not setting a certain threshold (such as 85%), but a percentage deviation from the baseline that would lead to the alarm. For example, Forra Wakidi et al. suggested a 5% decrease in SpO_2_ value within 5 s from the baseline as indicative of apnea [2].

The limitations of the device include the fact that the generalizable applicability is probably limited. Due to the psychological component, not all patients may benefit equally depending on the individual situation. More mentally impaired patients probably will not be able to correctly use the device, which needs to be correctly attached and at least a caregiving person is needed to adjust the app and send the data for analysis.

One downside of the available low-cost consumer devices may be lower accuracy. Regular dropouts were also detected during reading, probably due to a loss of skin contact with the sensor. One conclusion may therefore be that improvements of the sensors and a more reliable attachment avoiding the slipping off the sensor during the night might be required. However, previous research has shown that finger sensors are typically more reliable in terms of motion artifacts than other types of sensors [1,2].

The starting point of this evaluation was the aim of improving sleep behavior, including the time to fall asleep and length of sleep. The results show that the prevailing sleep disorder of the patient was significantly reduced with the device. Over time, this might also increase survival, which is strongly correlated with SpO_2_ levels in lung cancer patients [11].

## 5. Conclusions

Sleep disturbances can be precursors to serious health problems, as can problems with night terrors or sleep apnea. This study investigated whether the simple measurement of the time course of the concentration of oxygen in the blood is suitable for detecting sudden respiratory disturbances sufficiently quickly for a prompt and effective warning by a vibration alarm. Indeed, the described device reliably warned of disturbances in the oxygen concentration in the blood using a vibration alarm. Sleep disturbances with sudden awakening did not occur when the device was used. The patient benefited from the security gained in this way and slept much more peacefully. This study also substantiates the hypothesis that oxygen deficiency in the blood had been a major cause of insomnia in the patient.

## Figures and Tables

**Figure 1 geriatrics-07-00043-f001:**
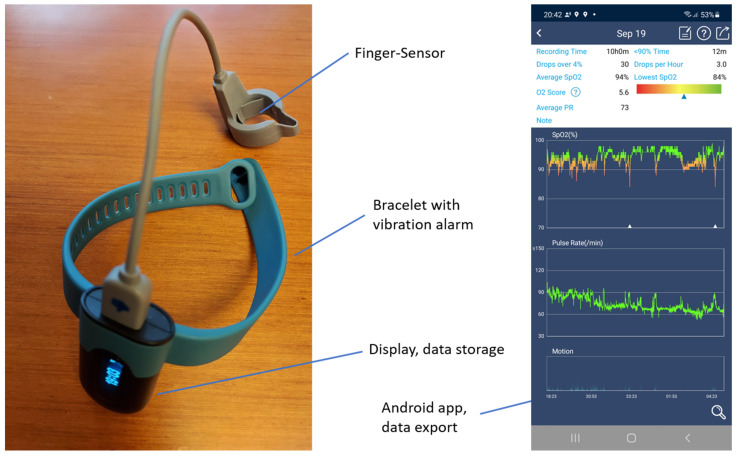
A picture of the device and a screenshot of the data evaluation in the android app.

**Figure 2 geriatrics-07-00043-f002:**
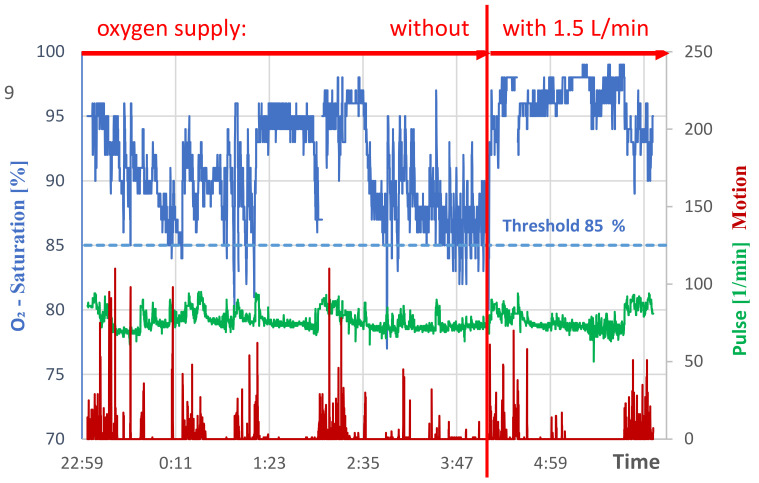
Night sleep of the patient with acute pneumonitis until shortly after 4:00 a.m. without oxygen supply, then with oxygen supply 1.5 L/min.

**Figure 3 geriatrics-07-00043-f003:**
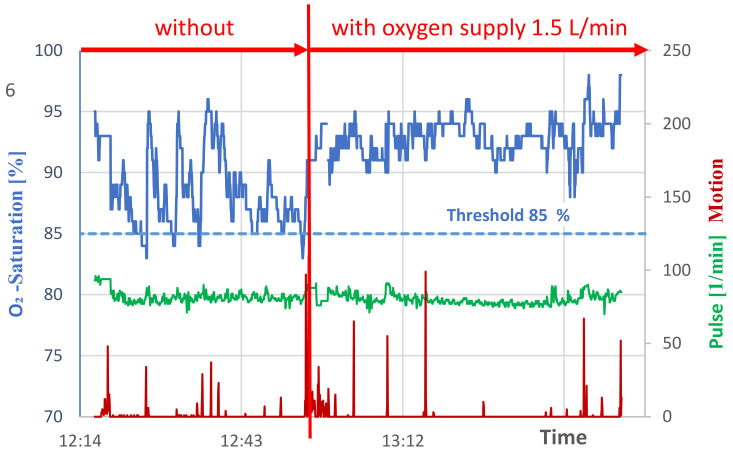
Midday nap of the patient with subsiding pneumonitis until shortly before 13:00 h without oxygen supply, then with oxygen supply 1.5 L/min.

**Figure 4 geriatrics-07-00043-f004:**
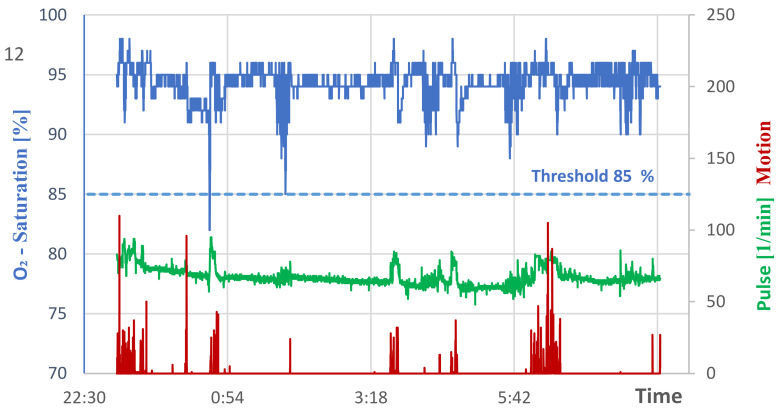
Night sleep of the almost-recovered patient without oxygen supply.

**Table 1 geriatrics-07-00043-t001:** Reaching and falling below the threshold value ^1^ of oxygen concentration in the blood during night sleep according to Figure 2.

Duration Sec.	Time from	to	Duration Sec.	Time from	to
4	0:08:31	0:08:35	12	3:44:31	3:44:43
48	0:15:55	0:16:43	20	3:46:27	3:46:47
8	0:50:47	0:50:55	32	3:49:27	3:49:59
48	0:55:55	0:56:43	32	3:51:31	3:52:03
12	0:59:43	0:59:55	32	3:54:27	3:54:59
44	1:05:27	1:06:11	4	3:55:03	3:55:07
32	1:11:31	1:12:03	4	3:58:07	3:58:11
24	2:53:43	2:54:07	32	3:58:15	3:58:47
12	2:57:31	2:57:43	8	4:01:55	4:02:03
16	3:01:59	3:02:15	20	4:03:59	4:04:19
8	3:02:39	3:02:47	16	4:07:55	4:08:11
20	3:40:47	3:41:07	36	4:11:31	4:12:07
4	3:41:35	3:41:39	12	4:12:15	4:12:27

^1^ Triggering of the vibration alarm at 85%. The duration of the undershoot of this threshold value by 1% is measured.

**Table 2 geriatrics-07-00043-t002:** Falling below the threshold value ^1^ of the oxygen concentration in the blood by 1% during the Midday sleep from Figure 3.

Duration Sec.	Time from	to
56	12:25:23	12:26:19
20	12:30:59	12:31:19
20	12:35:31	12:35:51
24	12:53:55	12:54:19

^1^ Triggering of the vibration alarm at 85%. The duration of the undershoot of this threshold value by 1% is measured.

**Table 3 geriatrics-07-00043-t003:** Falling below the threshold ^1^ of the oxygen concentration in the blood during night sleep from Figure 4.

Duration Sec.	Time from	to
28	0:36:33	0:37:01

^1^ Triggering of the vibration alarm at 85%. The duration of the undershoot of this threshold value by 1% is measured.

## Data Availability

All data generated or analyzed during this study are included in this published article (and its Appendix A).

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
