# Peer review of "Home Monitoring of Oxygen Saturation Using a Low-Cost Wearable Device with Haptic Feedback to Improve Sleep Quality in a Lung Cancer Patient: A Case Report"

_geriatrics, 2022, doi:10.3390/geriatrics7020043_

Round 1

Reviewer 1 Report

The authors report the case of 76-year-old man with lung carcinoma. The results are very promising with application SleepU device in lung cancer patients’ for the improvement their life quality.

I would like to ask the authors following questions:

  1. How the authors excluded sleep related breathing disorders? For this end is necessary Polysomnography not polygraphy.
  2. page 3, line -126, as I guess this device records the data on patient’s rest-activity pattern, but not sleep.
  3. page 4, line- 137, “alarm was set to go off at 85% oxygen saturation” or alarm was set to go on?
  4. page 7 line-247-251 – I am interested how was assessed sleep behavior improvement? How did you measure sleep latency and sleep duration?

Author Response

The authors report the case of 76-year-old man with lung carcinoma. The results are very promising with application SleepU device in lung cancer patients’ for the improvement their life quality.

REPONSE: Thank you for the positive assessment of our paper.

I would like to ask the authors following questions:

1. How the authors excluded sleep related breathing disorders? For this end is necessary Polysomnography not polygraphy.

RESPONSE: Because the polygraphy did not confirm the suspected diagnosis, there was no indication for polysomnography, which also involves very high costs for the diagnosis (around 5000 EUR) and would have involved a high level of inconvenience and stress for the geriatric patient. Therefore, is was decided not to conduct such a diagnosis. We have added a remark about that around line 80.

Furthermore, it should be noted that according to according to the manufacturer, the device may be also used in sleep labs [1-2]. The FDA approval states: “The Oxiband Pulse Oximeter is a wrist pulse oximeter indicated for use in measuring, displaying, storing, and transmitting functional oxygen saturation of arterial hemoglobin (SpO2) and pulse rate for adult patients. It is intended for spot-check and/or continuous data collection, and not continuous monitoring. It can be used in sleep labs, long-term care, hospitals and home use.”

The FDA registration document states non-clinical and clinical information provided by the manufacturer. The product was shown as equivalent to similar devices. The device is also “CE” approved and the CE label is printed on the backside of the device. Identification number 0197 (TÜV Rheinland, Germany).

References:

[1] https://getwellue.com/pages/sleepu-oxygen-monitor

2. page 3, line -126, as I guess this device records the data on patient’s rest-activity pattern, but not sleep.

RESPONSE: Yes, basically the sleep tracking is only indirect, as the major purpose of the device is SpO2 measurement. The sleep acitivity can be indirectly assessed from the motion sensor.

3. page 4, line- 137, “alarm was set to go off at 85% oxygen saturation” or alarm was set to go on?

RESPONSE: The wording was changed as requested for better clarity.

4. page 7 line-247-251 – I am interested how was assessed sleep behavior improvement? How did you measure sleep latency and sleep duration?

RESPONSE: The sleep quality was assessed by (i) the patient’ own subjective assessment and sleep protocols regarding start and end of sleep, and (ii) comparing the curves from motion sensor and pulse measurements of the device. The patient was also wearing a smartwatch that included sleep quality measurements (Withings Pulse HR). The sleep scores measured by the device were improved to “good” or “average” on most days in November 2021. However, we were unable to directly correlate the measurements as the patients started using the smartwatch only after the initial research period.

Reviewer 2 Report

The manuscript is well written with fluent English language. The objective of the study is to introduce the use of existing monitoring method for SpO2 with haptic feedback to reduce lung cancer patient’s anxiety and poor sleep.

Few concerns:

The last sentence in Introduction section lines 55-58 ,where you present the objective of this study is very unclear. You want to show impact of wearable oximetry with haptic feedback to help patient to handle his severe sleep disturbances or something more like that.

It leaves unclear to me how this oxygen supply was triggered in these examples? Do patient decide himself it is needed or caregiving person? When patient awake during night or nap, what  did he do? Did he check SpO2 values and try to breath calmly?

Patient is anyway wake up in all these examples and thus sleep is fragmented and this causes severe problems in longer run. The oxygen supply is necessary to give undisturbed sleep. Maybe it is better to use CPAP  device with oximetry instead?

Author Response

The manuscript is well written with fluent English language. The objective of the study is to introduce the use of existing monitoring method for SpO2 with haptic feedback to reduce lung cancer patient’s anxiety and poor sleep.

RESPONSE: Thank you for the kind assessment of our paper.

Few concerns:

The last sentence in Introduction section lines 55-58, where you present the objective of this study is very unclear. You want to show impact of wearable oximetry with haptic feedback to help patient to handle his severe sleep disturbances or something more like that.

RESPONSE: Thank you for suggesting to clarify the aim of the study. We have changed the text according to the suggestion of the reviewer.

It leaves unclear to me how this oxygen supply was triggered in these examples? Do patient decide himself it is needed or caregiving person? When patient awake during night or nap, what did he do? Did he check SpO2 values and try to breath calmly?

RESPONSE: The oxygen supply was manually triggered. There was no connection between the oxygen generator and the wearable device. When the device triggered several times, and the patient did have problems breathing even after waking up, the patient started the oxygen supply himself. We have clarified that this was a manual oxygen setup in section 2.1.

Patient is anyway wake up in all these examples and thus sleep is fragmented and this causes severe problems in longer run. The oxygen supply is necessary to give undisturbed sleep. Maybe it is better to use CPAP device with oximetry instead?

RESPONSE: The patient had an extremely low tolerance of the CPAP mask during sleeping and it increased his feeling of anxiety, so that he had problems sleeping with the inconvenience of the mask as well. He barely tolerated the nasal oxygenation. Therefore, the therapy goal was to avoid oxygenation or CPAP if possible. According to the patient’s report, despite the fragmentation of the sleep due to the waking by the device, his individually perceived sleep quality was better than in nights under oxygenation. When woken by the vibration alarm, he was usually able to quickly sleep again.

This manuscript is a resubmission of an earlier submission. The following is a list of the peer review reports and author responses from that submission.

Round 1

Reviewer 1 Report

This paper describes a clinical application of low-cost pulse oximeter. As a case report, this paper is good for clinical practice, but there are no scientific interests for biosensor engineers, Because the authors presented common results which simply observed the oxygen saturation with and without oxygen supply. If the authors tried an automated oxygen supply from the output of pulse oximeter, the total system is very interesting to respiratory scientists.

This paper will be submitted to other appropriate journals. The device may be promising use of clinical practice with easy to use, low-priced, and easy data collection. Also, this paper is a good article in clinical practice.

Major

In detail of 2.2, the presentation is only a technical note. The accuracy in case of motion artifact and long-term stability must be presented.

Reference 10 is not a peer-reviewed paper and it is not suitable for reference.

The results are only observation reports. If the authors claim the sleep quality, the continuous pulse rates are calculated and the ratio of sympathetic and parasympathetic activities can be estimated. The authors got the CSV data and the stress index can easily be obtained.

Reviewer 2 Report

This paper reports a case of a lung cancer patient with increasing difficulties in falling asleep and frequent periods of wakefulness. The authors hypothesized that the symptoms described were causally related to a drop in oxygen saturation in the patient's blood. They used a low-cost wearable device with haptic feedback to home monitor oxygen saturation for improving sleep quality in a lung cancer patient. The quality of this paper is low. They only use one person’s data to prove a medical hypothesis. The results don’t have statistical significance. They should clarify the novelty of this paper.